# Reproducibility report of "Mind the pad"

## Reproducibility Summary

**Scope of Reproducibility**

Convolution mechanism is widely adopted for a large variety of tasks like image classification or object detection. Alsallakah et al. [1] demonstrate how this mechanism has some flaws caused by padding. Our aim is to reproduce the following results

- Single Shot Detector blind spot on small object detection
- Single Shot Detector blind spot fix when changing padding mode from *zeros* to *reflect*
- Uneven application of padding on downsampling convolutional layers causes feature map erosion and lower accuracy.

Once reproduced these results, we performed a series of ablation studies to understand the effect of related factors in a CNN

- How does Batch Normalization interact with uneven application of padding?
- Which category between these: 1) padding modalities {*zeros, reflect, circular, replicate*} 2) with/without batch normalization 3) with/without uneven application of padding is more shift robust on image classification?

**Methodology**

To reproduce paper claims we implemented all the experiments from scratch in order to have more reliable results. The only external resource used in this article is a PyTorch implementation of Single Shot Detector made by NVIDIA [6]. Furthermore, since the paper thesis is related not to a specific implementation but to a class of models, our implementations fit inside the same category but with a different configuration. We have done so to stress paper claims and to confirm their general validity.

To train the models for image classification, 48, we used a local Nvidia GeForce GTX 1660 with 6 GB of memory, 8 GB of RAM, and AMD Ryzen 5 2600X (12) @ 3.600GHz. The training holds for approximately 20 hours. The reason to train such quantity of models are: 1) 4 padding modes 2) with/without Batch Normalization 3) with/without uneven application of padding (i.e. input images zero/one padded) 4) 3 random seeds

**Results**

During our experiments, we found the same blind spots of paper authors but on a different location: for us, the blind spot was close to the right border while paper claims it is at the top border. We fixed blind spot issue using *reflect* padding instead of *zeros* like paper does. About uneven application of padding, we have a performance improvement when comparing models with/without uneven application of padding but with different delta. In our experiments, the accuracy improvement is around 0.8% while averagely of 0.4 % for the original paper. We believe that the cause is a different model architecture and dataset. In this article, we adopted a simple Sequential CNN classifier on Letter MNIST while the paper uses famous architectures like ResNet on Imagenet.

Submitted to ML Reproducibility Challenge 2020. Do not distribute.

**What was easy**

- Obtain paper results about image classifier uneven application of padding worked at the first shot.

- Once found good implementation of SSD reproduce results on evaluation was immediate.

**What was difficult**

- Find an image classifier architecture suitable for uneven application of padding tests like the article i.e. with original input size the downsampling layers don't see the right padded border while with one-pad images the downsampling layers see all padded borders.

- Change padding mode of convolutional layers or disable Batch Normalization layers especially on Tensorflow models.

- Understand how to plot SSD's object confidence for all zones of the image

**Communication with original authors**

To make this article there wasn't any contact with the original authors.

# 1 Introduction

Convolution algorithm is adopted in a large variety of tasks, such as image classification, object detection, and generative models. However, despite its diffusion, there are a lot of hidden mechanisms we don't fully understand about convolution. For instance, Alsallakah et al. [1] noted that an object detector, SSD [5], fails to recognize traffic lights on consecutive frames while previously the object was identified. The major difference between frames was a vertical shift of the traffic light. Starting with this observation, they demonstrate how padding can create blind spots on object detection tasks. Furthermore, on image classification, they shown how downsampling layers (convolutions with stride > 1) don't see the right border leads to lower accuracy. They also proposed a series of constrains to avoid that phenomena, called by them uneven application of padding.

The article focuses on reproducing the same results of Alsallakah et al. [1] through different datasets and models but in the same task category. For example, instead of detecting blind spots on traffic light [2] detection with SSD [5], we tested Alsallakah et al. [1] blind spot thesis using COCO dataset on images containing only small items. In the second part, we trained an image classifier from scratch on Letter MNIST to verify if the uneven application of padding leads to feature erosion. Furthermore, we provided a set of ablation studies to evaluate the effectiveness of Batch Normalization and tested shift robustness.

# 2 Object Detection with SSD

## 2.1 Introduction

Single Shot Detector [5] (SSD) is an object detection model, able to identify items of different scale and size inside the image in a single step. This allows the model to be easily deployable in a real time environment.

In our experiments, we used SSD [6] trained on COCO instead of traffic light [2], as Alsallakah et al. [1] does. For evaluation we took traffic light [2] images like Figure 2 from COCO to fit in the same category of Alsallakah et al. [1] , i.e. small object detection.

## 2.2 Results

Alsallakah et al. [1] demonstrate how padding allows the model to exploit border locations causing identification problems when close to the borders. Figure 3 shows very similar results of the original paper. The blind spot differs just for the location: in SSD [5] trained on COCO the blind spot is close to the right border while for the original authors is close to the top border.

We found coherent results when changing padding mode from *zeros* to *reflect*, as Figure 3 shows. We tried as well padding mode *circular* but it isn't very effective to counter blind spots on SSD [5] evaluation.

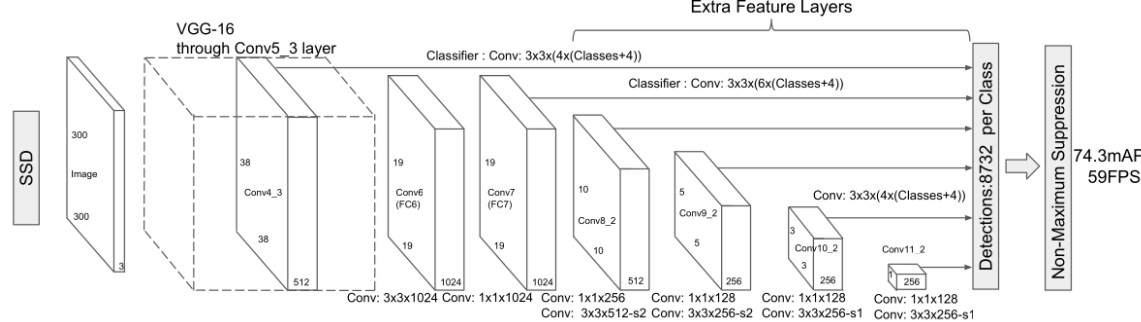

Figure 1: Single Shot Detector architecture taken from [5]. Our version has MobileNet [4] instead of VGG

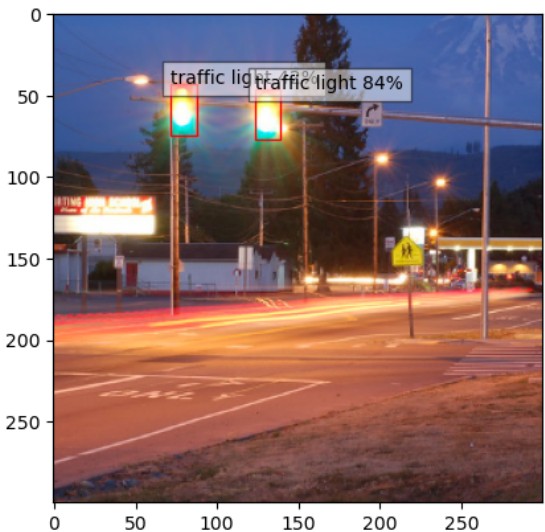

Figure 2: Sample of SSD [5] object detection with confidence above each bounding box

## 3 Spatial bias on image classifiers

### 3.1 Experiment setup

In order to check if Alsallakah et al. [1] thesis is correct, we recreated the same experiment condition but with different models and datasets.

**Dataset**    Dataset is Letter MNIST, a set of grayscale {28, 28} images showing handwritten alphabet letters with 26 different classes. Train split has 60 000 samples while test split has 10 000.

**Model**    We trained a set of small CNN feature extractors followed by Average Pooling and two Fully Connected layers. Every convolutional layer has "same" padding and downsample every two (Figure 5). Given this architecture, cardinal design choices are:

- Average pooling after Convolutional layers instead Max pooling because it preserves shift equivariance property and therefore, results are more reliable.
- Same padding because it's the most common in CNN. It allows the model to preserve the image size through convolutional layers. This is essential for architectures like ResNet [3] or to handle variable input size.

We followed the same Alsallakah et al. [1] uneven application of padding (Figure 6) experiments idea. The model architecture is chosen such that downsampling layers see all padded borders (no uneven application of padding) when

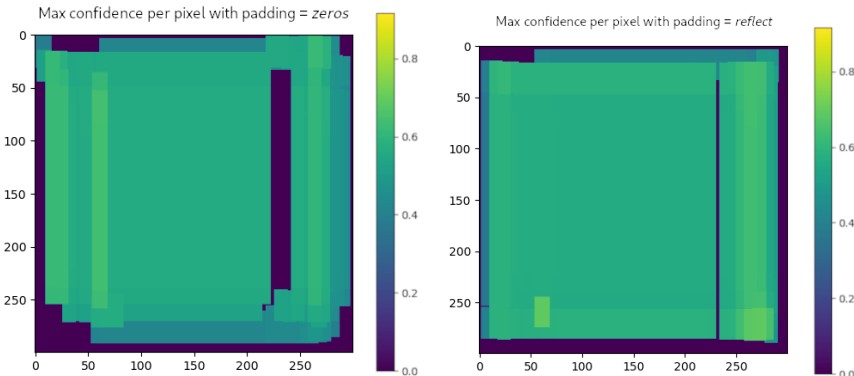

Figure 3: Example of blind spot on Figure 2 obtained by moving the left semaphore through all the image. At evaluation time SSD [5] has a blind spot close to the right border while Alsallakah et al. [1] found that on top border

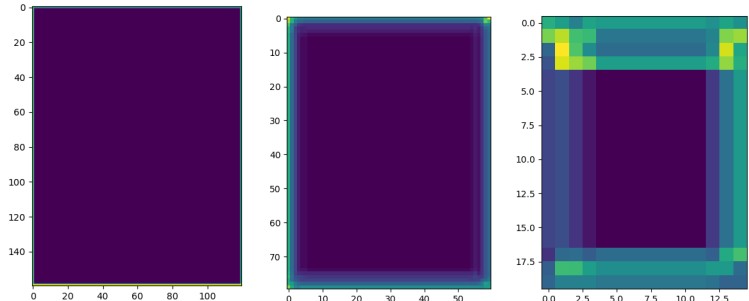

Figure 4: SSD ReLU output when feed a black image. From the left to the right, we have respectively ReLU outputs of 2nd, 9th and 19th layers. It is clear how going deeper through layers worsen border bias.

input size is $\{29, 29\}$ i.e. padded one. Instead with the original input size $\{28, 28\}$, downsampling layers don't see the right padded border leading to feature maps erosion.

**Training procedure**   We have chosen the simpler training possible: model parameter optimization continues until validation loss decreases (Early stopping) using Adam optimizer with learning rate $10^{-3}$.

## 3.2   Accuracy of models without uneven application of padding

To validate paper results, we do not aim to have the same absolute value of accuracy because model architecture, dataset, and the number of classes differ. But rather we aimed to the same relative accuracy improvement when fixing the problem of uneven application of padding.

As Table 1 shows, we have slightly better results to Alsallakah et al. [1] in terms of relative accuracy improvement. When uneven application of padding is missing, accuracy improves averagely of 0.8%. We believe that our improvements are slightly higher because we have done the same experiments with a simpler dataset and model architecture compared to the original paper. In their results, they showed various versions of ResNet [3] and MobileNet [4] trained on ImageNet [7].

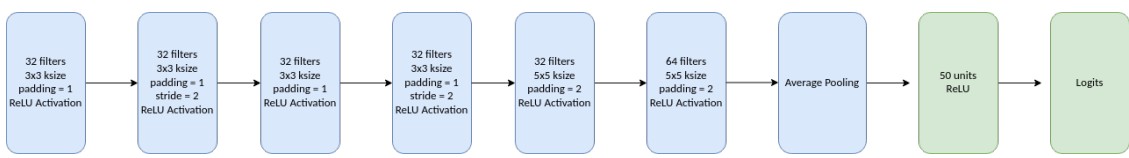

Figure 5: CNN classifier with same padding used in Section

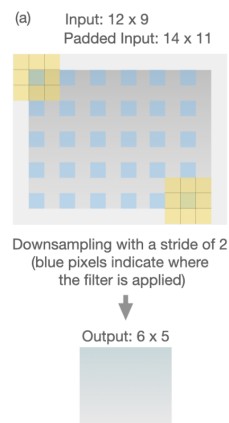

Figure 6: Example of uneven application of padding. Courtesy of Alsallakah et al. [1]

| Padding mode / Input size | circular | reflect | replicate | zeros | Marginal |
|---|---|---|---|---|---|
| {29, 29} | $92.7 \pm 0.2\,\%$ | $93.1 \pm 0.2\,\%$ | $93.0 \pm 0.3\,\%$ | $93.0 \pm 0.2\,\%$ | $93.0 \pm 0.2\,\%$ |
| {28, 28} | $92.0 \pm 0.3\,\%$ | $92.3 \pm 0.3\,\%$ | $92.2 \pm 0.3\,\%$ | $92.1 \pm 0.3\,\%$ | $92.2 \pm 0.3\,\%$ |

Table 1: Accuracy of CNN classifier by padding mode and presence of border asymmetry in downsampling layers. Metrics results are in the form $\mu \pm \sigma$ because for each cell in the table we trained 3 identical models but with different random seed in order to prevent the influence of aleatory factors like batch sampling or parameter initialization

Instead, we haven't seen big accuracy variations with any type of padding mode. Only padding mode *circular* is slightly worse in terms of accuracy compared to the others, around $0.3\%$. We tested shift robustness of models with any padding mode, resulting in *circular* much more robust than others. Further details are in ablation studies section.

**Test only uneven application of padding** We check if Alsallakah et al. [1] uneven application of padding thesis holds only at evaluation time. We test models trained with uneven application of padding without it and vice-versa.

| Test input size / Training input size | {28, 28} | {29, 29} |
|---|---|---|
| {29, 29} | $92.0 \pm 0.2\,\%$ | $93.0 \pm 0.2\,\%$ |
| {28, 28} | $92.1 \pm 0.3\,\%$ | $72.9 \pm 2.4\,\%$ |

Table 2: Accuracy of CNN classifier on Letter MNIST fed with border asymmetry images (size {28, 28})

Table 2 shows a performance decrease from 92.1% to 72.9% when the model trained with the uneven application of padding is evaluated without it. This result eliminates the hypothesis that just uneven application of padding leads to performance decrease. Rather, the true cause is the uneven application of padding training leads the model to learn skewed filters and therefore to worse accuracy. Furthermore, it confirms Alsallakah et al. [1] binding between asymmetrical filters and performance decrease.

| Training Input size With batch norm | {28, 28} | {29, 29} | Marginal |
|---|---|---|---|
| True | 14.612547 | 13.811748 | 14.212148 |
| False | 15.558550 | 14.280053 | 14.919302 |
| Marginal | 15.085549 | 14.045901 | |

Table 3: Perplexity of CNN classifiers with zero/one pad and with/without Batch Normalization. Model with Batch Normalization and one padded tends to have less perplexity at test time

## 3.3  Code to check if model has uneven application of padding

The following python function tells if a model with a fixed input size has uneven application of padding on downsampling layers. This implementation uses the same equations of analyzed paper Section 5. It supports *PyTorch* models and leverages on the package *Torchinfo*, which tells input and output size of each CNN's layer.

```python
import torch.nn as nn
from torchinfo import summary

def has_uneven_padding(model: nn.Module, input_size: tuple):
    summary_info = summary(model, input_size, verbose=0)
    print(summary_info)
    for layer_info in summary_info.summary_list:
        if isinstance(layer_info.module, nn.Conv2d):
            conv_module = layer_info.module
            if any(s > 1 for s in conv_module.stride):  #downsampling layers
                *_, h_i1, w_i1 = layer_info.input_size
                *_, h_i, w_i = layer_info.output_size
                h_i_line = h_i1 + 2 * conv_module.padding[0]
                w_i_line = w_i1 + 2 * conv_module.padding[1]

                h_i_hat = conv_module.stride[0] * (h_i - 1) + conv_module.kernel_size[0]
                w_i_hat = conv_module.stride[1] * (w_i - 1) + conv_module.kernel_size[1]

                if h_i_line != h_i_hat or w_i_line != w_i_hat:
                    print('layer', layer_info.var_name)
                    print(f'{h_i_line = }', f'{h_i_hat = }', f'{w_i_line = }', f'{w_i_hat = }')
                    return True
    return False
```

# 4  Ablation studies

## 4.1  Batch Normalization impact

Once asserted coherent results with Alsallakah et al. [1] , we evaluate the impact of other essential CNNs components like Batch Normalization. Using the same CNN classifier of Section 3, we added Batch Normalization after every Convolutional layer.

We **haven't observed a significant difference in terms of accuracy** of the model with Batch Normalization. Still, they tend to have lower perplexity than models without Batch Normalization as we can see in Table 3.

## 4.2  Shift robustness and prediction error

Following Alsallakah et al. [1] paper core principles, we analyzed *spatial bias* of image classifiers through shift robustness. With the latter concept we intend to pose the following question: the model is able to classify correctly the image even if input is shifted towards the borders?

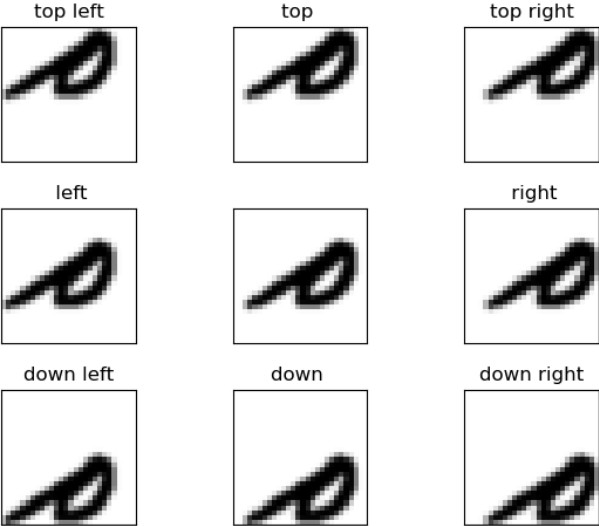

(a) Example of shifted letter "$a$" where the original one is on the center. Shifts are computed algorithmically such that they push the letter at one border between $\{top, left, down, right\}$

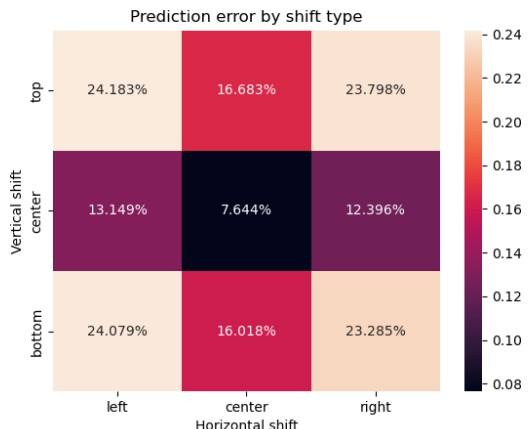

(b) Prediction error when image is shifted toward borders. Average error is simmetric with regards to the center, except for a slight higher error when shifted to the left instead of right.

**Shift methodology** To generate a shifted dataset, we pushed the letter into border through any combination of these directions $\{north, center, south\} \times \{left, center, right\}$ like in Figure 7a. Therefore, we evaluated the shifted images with the trained models of Section 3 and measured how the prediction error changes when shifting the letter toward border, as Figure 7b shows.

**Results** After shifting in 9 directions 10 images per class we computed prediction error by models with/without Batch Normalization and with/without uneven application of padding (Figure 8). We found that models without uneven application of padding and with Batch Normalization are slightly more shift robust than other categories. Furthermore, when horizontal shift is very large, for instance with letters like *l*, all models almost random guess the label.

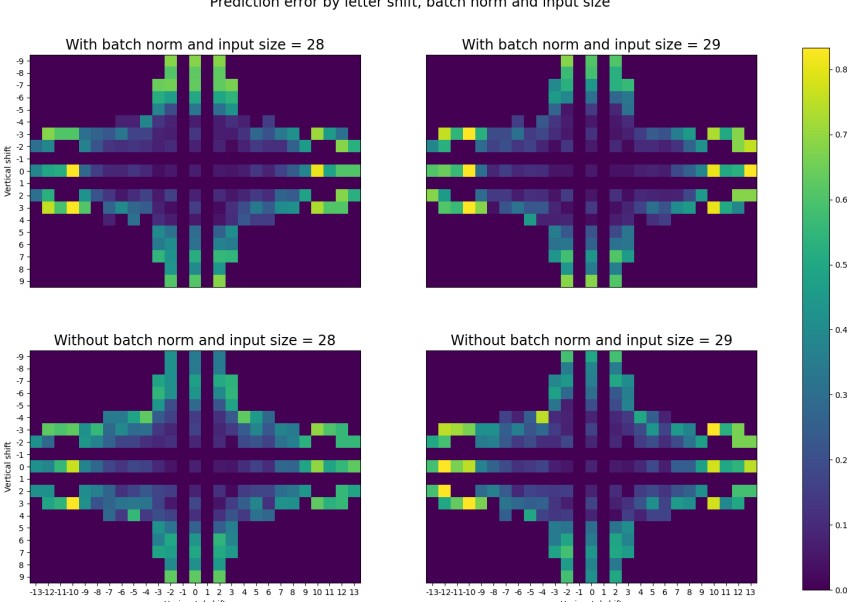

Figure 8: Prediction error by models with or without Batch Normalization, zero or one input padded. Each heatmap shows the prediction error got by shifting images toward borders as Figure 7a. Heatmap values follows a "+" pattern because letters are exclusively big vertically or horizontally. Results show that models with Batch Normalization and one pad are slightly more robust than the others.

Therefore we evaluated shift robustness by padding modes (Figure 9). Interestingly, models with padding mode *circular* eliminate almost all shift biases except when horizontal shift is extremely high.

We speculate that the primary reason for *circular* padding shift robustness is because it takes padded values from the opposite side of the feature map. By taking very distant values, the padded feature map has a higher degree of variance compared to other methods like *zeros* of *reflect* padding modes, resulting in a less "exploitable" border.

## 5 Conclusion

Thanks to the clarity of the paper we managed to reproduce all the main paper results with other model architectures and datasets, which makes their claims more robust. In detail, our reproduction procedures are

1. We evaluated Single Shot Detector (SSD) [5] trained on COCO dataset instead of traffic light [2] dataset obtaining compatible results. By changing SSD padding mode from *zeros* to *reflect* we fixed blind spot issue as they do.

2. We checked if image classifiers suffers from uneven application of padding using a CNN classifier built from scratch instead of using more famous architectures like ResNet [3] and MobileNet [4]. We found the same accuracy improvement when border asymmetry is removed by downsampling layers.

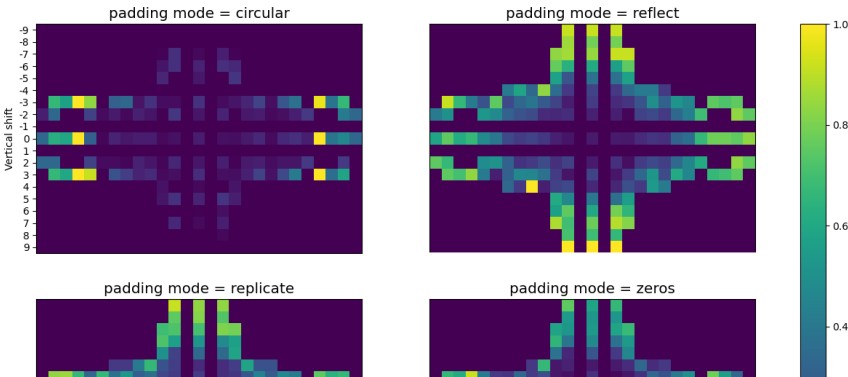

Figure 9: Prediction error by models trained with padding mode {*zeros, replicate, reflect, circular*}. Each plot shows prediction error got by shifting images toward borders like Figure 7a. Clearly, padding mode *circular* is much more shift robust than other modalities.

Following, we performed a series of ablation studies which results are

- Batch Normalization doesn't improve accuracy without uneven application of padding (Figure 6). We observed just a slight improvement in terms of perplexity.

- We evaluated shift robustness of image classifiers by shifting the image content towards the borders through a combination of the cardinal directions *{north, south, east, west}* (Figure 7a). In this context, we found that models with padding mode *circular* are much more shift robust than others (Figure 9) but with a slight decrease of average accuracy ($-0.3\%$)

To conclude our Reproducibility report, we propose some directions to improve actual results:

- Research new padding modalities such that they are very efficient during training and prevents blind spots on Object detection.
- Develop new algorithms to prevent at test time skewed filters without worsening training computational cost.

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
