# OpenReview forum: "Reproducibility report of "Mind the pad""
_ML_Reproducibility_Challenge/2021/Fall — Reject_

### Official Review · Reviewer_sUTZ · 2022-03-08
**Reproducibility report of "Mind the pad"**

**Rating:** 3
**Confidence:** 4

**Review:**

This paper (Re-Paper) aims to reproduce the the padding bias as presented in this paper "https://openreview.net/pdf?id=m1CD7tPubNy" (Ori-Paper).
Instead of following the flow of the Ori-Paper, the Re-Paper evaluated the padding bias on different network architectures and different datasets. Some of the conclusions from the Ori-Paper are re-confirmed, such as how paddings bias have causing issues close to the borders and how padding bias can be detrimental to small object detection tasks. The Re-Paper also evaluated the scenario when the uneven padding is not consistently applied in training and testing and re-confirmed that uneven padding could cause asymmetric mean filters and thus cause accuracy drop. In addition to re-confirming the conclusions from the Ori-Paper, the Re-Paper also tested the impact of uneven padding on batch normalization and did not find significant impact.

Overall, this Re-Paper provided empirical results in an un-organized manner. The Re-Paper is not easy to read and follow, which needs to be restructured to be more readable. Because the implementation did not reproduce the results in the Ori-Paper, it is unclear whether the implementation is flawless. In particular, in the Table 2, the accuracy drop from 92%-93% to about 73% is huge. It's unclear whether uneven padding is the only cause. The additional test on batch normalization is lack of rationale. It is unclear why padding bias may lead to impact on batch normalization.

---

### Official Review · Reviewer_L5zc · 2022-03-09
**Does not add enough value regarding the reproducibility of the original paper**

**Rating:** 5
**Confidence:** 3

**Review:**

**Reproducibility Summary**

Yes, the authors have included a reproducibility summary on the first page, as part of the template. The summary is well written and covers their major findings. However, there is an issue that their summary extends well over 1 page, while it was mandatory to have the summary fit in one page.


**Scope of reproducibility**

Yes, the report clearly states the scope of reproducibility, in the sense that they would not replicate the exact results in the original paper. They aim to test the main hypothesis of the original paper (padding may create blind spots) on different datasets.


**Code**

The authors did not use the original paper’s code, rather implemented it all from scratch. Since they experimented on different datasets and models than the original paper, working on the original paper’s code repository would not have been of much use.

**Communication with original authors**

The authors did not contact the original authors at any point for conducting this reproducibility study

**Hyperparameter Search**

The authors do not use the original paper’s code to reproduce the results, and also do not experiment on the same dataset and model from the original paper. Hence, comparing their hyperparameter search with that in the original paper does not have much merit.

**Ablation Study**

The report provides new ablation studies, where they study the effect of batch normalization with an uneven application of padding. Such ablations were not part of the original paper.


**Discussion on results**

The report works with entirely different datasets and models as compared to the original paper, hence a lot cannot be inferred about the state of reproducibility of the original paper. Authors state that they find a similar phenomenon occurring on a different dataset (COCO instead of traffic light detector) with a different model (custom CNN trained on Letter MNIST instead of ResNet trained on ImageNet). But the report does not reproduce the results in the original paper on the traffic light detector dataset, hence we cannot conclude much about what parts of the original paper were easy/hard to reproduce from this report.

**Recommendations for reproducibility**

As stated in the previous section, since the authors work with a different dataset and model as compared to the original paper, it would be challenging to provide any recommendations that could be used for improving the reproducibility of the original paper.

**Results beyond the paper**

The report does an excellent job in this aspect, as they verify the main claim of the original paper on a new dataset (COCO) and models (custom CNN trained on Letter MNIST). Also, they study the effect of batch normalization, which was not done in the original work.

**Overall organization and clarity**

There are several grammatical issues, to mention a few, lines 34, 35 in the reproducibility summary. Line 10 has a typo, it should be “Once we reproduced these results”. Line 162 should be rephrased as follows: “... to prevent skewed filters at test time without … ”. Line 133 should be rephrased as follows: “ … as shown in Figure 7b”. Line 91 seems to have a typo, it should be “simplest” I guess. Line 52 has a typo, it should be “constraints“. Line 58 has a typo, it should be “we provide a set of ablation studies”.

**Summary**

While the authors have some interesting results that the hypothesis of the original paper holds on new datasets, I feel enough discussion was not present around the reproducibility of the main results of the original paper. Also, there are issues with the presentation of the paper, the reproducibility summary exceeds one page and there are plenty of grammatical errors in writing. Hence, I feel the reproducibility report in its current form might not be ready.

---

### Meta-Review · Area_Chair_fpw3 · 2022-04-08

**Recommendation:** Reject
**Confidence:** 4

**Metareview:**

This paper attempts to reproduce the results reported in the original paper. However, as the reviewers pointed out, the results reported don't completely verify the claims. Moreover, there are several presentation and writing issues and hence the paper is not ready for acceptance.

---

### Decision · Program_Chairs · 2022-04-09

Reject